# (2-Hydroxy-3-Methoxybenzylidene)thiazolo[3,2-*a*]pyrimidines: Synthesis, Self-Assembly in the Crystalline Phase and Cytotoxic Activity

**DOI:** 10.3390/ijms24032084

**Published:** 2023-01-20

**Authors:** Artem S. Agarkov, Anna A. Nefedova, Elina R. Gabitova, Dilyara O. Mingazhetdinova, Alexander S. Ovsyannikov, Daut R. Islamov, Syumbelya K. Amerhanova, Anna P. Lyubina, Alexandra D. Voloshina, Igor A. Litvinov, Svetlana E. Solovieva, Igor S. Antipin

**Affiliations:** 1Arbuzov Institute of Organic and Physical Chemistry, FRC Kazan Scientific Center, Russian Academy of Sciences, Arbuzova 8, 420088 Kazan, Russia; 2A.M. Butlerov Chemical Institute, Kazan Federal University, 18 Kremlevskaya St., 420008 Kazan, Russia; 3Laboratory for Structural Studies of Biomacromolecules, FRC Kazan Scientific Center of RAS, 2/31 Lobachevskogo Str., 420111 Kazan, Russia

**Keywords:** thiazolo[3,2-*a*]pyrimidines, crystal structure, non-covalent interactions, halogen and hydrogen bonding, chiral discrimination, cytotoxicity, antitumor agents

## Abstract

A series of new 2-hydroxy-3-methoxybenzylidenethiazolo[3,2-*a*]pyrimidines with different aryl substituents at the 5 position are synthesized and characterized by ^1^H/ ^13^C NMR and IR-spectroscopy and mass-spectrometry, as well as single crystal X-ray diffraction (SCXRD). It was demonstrated that the type of hydrogen bonding can play a key role in the chiral discrimination of these compounds in the crystalline phase. The hydrogen bond of the O–H...N type leads to 1D supramolecular heterochiral chains or conglomerate crystallization in the case of the formation of homochiral chains. The hydrogen bond of O–H...O type gave racemic dimers, which are packed into 2D supramolecular layers with a parallel or angular dimers arrangement. Halogen bonding of the N...Br or O...Br type brings a new motif into supramolecular self-assembly in the crystalline phase: the formation of 1D supramolecular homochiral chains instead 2D supramolecular layers. The study of cytotoxicity against various tumor cells in vitro was carried out. It was found that 2-hydroxy−3-methoxybenzylidenethiazolo[3,2-*a*]pyrimidines with 3-nitrophenyl substituent at C5 carbon atom demonstrated a high efficiency against M-HeLa (cervical adenocarcinoma) and low cytotoxicity against normal liver cells.

## 1. Introduction

Recently, great attention has been paid to thiazolo[3,2-*a*]pyrimidine derivatives due to their high biological activity. Thiazolo[3,2-*a*]pyrimidines have been proposed as potent GluN2A-selective *N*-methyl-*D*-aspartate receptor positive allosteric modulators (Figure 1A) [1], and are promising acetylcholinesterase (AChE) inhibitors (Figure 1B) [2,3]. Some of these compounds possess a powerful antileishmanial effect (in vitro) against the forms of promastigote (Figure 1C–E) [4], as well as high cytotoxicity and good selectivity against the MCF−7 cell line (Figure 1F,G) [5,6]. In addition to antitumor activity, thiazolo[3,2-*a*]pyrimidine derivatives can exhibit high antibacterial, antidiabetic and antifungal activity (Figure 1H–L) [7,8,9].

One of the promising synthetic ways to create highly effective drugs is a combination of thiazolopyrimidines with various pharmacophore frameworks into new hybrid structures. For example, a thiazolo[3,2-*a*]pyrimidine system modified by the acyclic monosaccharide moiety (acyclic analogues of C-nucleosides) (Figure 1M) demonstrated a high cytotoxicity against MCF 7 cancer cells and the Caco−2 cell line [10], whereas derivatives containing thiazolopyrimidine and hydrazone fragments did not show the expected antitumor effect [11].

The 2-arylmethylidene derivatives of thiazolo[3,2-*a*]pyrimidine with a 2- and 4-hydroxybenzylidene fragment (Figure 1N–Q) have exhibited high or moderate activity against a number of cancer lines of various genesis and moderate cytotoxicity against normal liver cells [12]. It was found out that they exhibit high or moderate activity against a number of cancer lines of various genesis, including cervical cancer cell lines (M-HeLa) and human duodenal adenocarcinoma (HuTu 80), with moderate cytotoxicity against normal liver cells. The derivative containing the 2-hydroxybenzylidene fragment demonstrated a particular efficiency and selectivity of cytotoxic effect against tumor cells in comparison with the commercial drug *Sorafenib*.

Since these derivatives have an asymmetric carbon atom, one of the most important problems is the separation of racemic mixtures into enantiopure isomers. Only one successful attempt to separate enantiomers of this class of compounds (Figure 1R) by chiral HPLC (WHELK-O column) has been reported [13]. It was found that enantiomers were stable and did not undergo racemization for more than one week at room temperature. Another strategy for the separation of racemic mixtures is based on the enantiomer chiral discrimination in the crystalline phase when the chiral molecules demonstrate a greater bonding affinity with the one enantiomer, leading to the conglomerate crystallization.

It is commonly known that different enantiomers can interact differently with chiral biomolecular receptors, leading to only one of the two enantiomers exerting the desired medicinal effect. Thus, the study aimed at searching for the proper method for enantiomer separation is of the pivotal importance because it can shed light on the biological activity mechanism of this class of macrocycles and potentially improve it while minimizing the side effect on the normal cells [14].

Quite recently, we showed that the supramolecular motif of (2-/4-hydroxybenzylidene)thiazolo[3,2-*a*]pyrimidines in the crystalline phase was controlled by intermolecular non-covalent interactions, mainly, hydrogen and halogen bonding [12]. Crystallization conditions leading to the formation of chiral supramolecular ensembles (Zonke space group), in particular, homochiral chains consisting of molecules of only one stereoisomer, were found. It was also demonstrated that Br–π, O–π and π–π interactions also play a significant role upon crystal packing [12,15].

Herein, we report on the synthesis and supramolecular organization in the solid phase and the anti-tumor activity of new 2-arylmethylidenthiazolo[3,2-*a*]pyrimidine derivatives **1–6**, bearing 2-hydroxy- and 3-methoxy groups in the benzylidene fragment. We expected that the introduction of a methoxy group can provide an additional coordination center to enhance chiral discrimination in crystals and bioactivity due to the effect of new intermolecular interactions.

## 2. Results and Discussion

Arylmethylidenethiazolopyrimidines **1–6** were synthesized according to Figure 1. In the first stage, a three-component Biginelli condensation of aldehyde, thiourea and a 1,3-dicarbonyl compound (acetoacetic ether or acetylacetone) in a molar ratio 1:1.5:1 led to the formation of 1,2,3,4-tetrahydropyrimidine−2-thion derivatives. The reaction was carried out under three different conditions in dependence of the nature of the 1,3-dicarbonyl compound and the aromatic aldehyde to achieve maximal yields for all target compounds. In the case of aromatic aldehydes containing electron donating substituents (4-methyl- and 2-methoxybenzaldehydes), the reaction can be effectively performed in the presence of a catalytic amount of iodine (0.03 mol) in the boiling acetonitrile. It was noted that electron withdrawing groups, such as 3-nitro-, 3- or 4-bromo in benzaldehydes, reduce the reactivity of these compounds in Biginelli reaction. Fusion of the initial reagents at 120 °C in the absence of a solvent allows us to achieve practically quantitative yields of the products. It was determined that the optimal method for the interaction between thiourea, ethyl acetoacetate and benzaldehyde was a reaction at solvent-free conditions carried out in the presence of concentrated sulfuric acid. In all cases, the yields were found to be in the range of 90–97%.

Thiazolo[3,2-*a*]pyrimidines–synthetic precursors of the target 2-arylmethylidene derivatives of thiazolopyrimidine were synthesized by the interaction of the obtained 1,2,3,4-tetrahydropyrimidine−2 with ethyl chloroacetate at 120 °C. The final condensation of thiazolopyrimidines with aromatic aldehydes was performed in ethanol in the presence of pyrrolidine as a base, producing excellent yields (95–98%) [16].

The structures of the synthesized compounds were studied using SCXRD, for which the data are in agreement with the structure of these compounds in a solution (for ^1^H-,^13^C-NMR-, IR- and mass-spectra; see Appendix A). In the crystalline phase, all the compounds obtained, as previously shown [12], have a similar structure when all carbon atoms, with the exception of C5, as well as heteroatoms (S and N) of the bicyclic fragment of thiazolo[3,2-*a*]pyrimidines are located in the same plane. All the obtained molecules exhibit only one C = C double bond configuration when the 2-hydroxy−3-methoxybenzylidene fragment and the S atom of the thiazolyl part are cis-oriented (Z-isomers).

Both of the distances observed for the C2–C9 and C9–C10 bonds, as well as the dihedral angle formed between the phenolic moiety of benzylidene substituent at the C2 atom and the thiazolo[3,2-*a*]pyrimidine bicyclic ring, are the evidences of a large electron conjugated system for all of the compounds’ formation (see Appendix A). In the crystalline phase, all of the studied 2-hydroxy−3-methoxybenzylidenethiazolo[3,2-*a*]pyrimidines adopt conformation when 2-OH and 3-OMe groups are found to be *syn* orientated with the carbonyl group of thiazolyl moiety.

For all compounds **1**–**6**, the methyl group in the arylidene substituent is orientated outwards, respecting the hydroxyl group. For both compounds **1** and **2**, the self-assembly in the crystalline phase produced the formation of supramolecular dimers structures composed of two enantiomers connected to each other via two intermolecular hydrogen bonds of the O–H…O type (Figure 2). The hydroxyl and methoxy groups of one isomer interact with the carbonyl O-atom of the thiazolidine fragment of another isomer (d_O3…O11_ = 2.810(1) Å for 1, d_O3…O11_ = 2.738(2) Å for 2).

In contrast to earlier reported analogous compounds [12], the formation of H-bonded dimers was observed independently of the nature of the used solvent (dimethyl sulfoxide or alcohol) upon the crystallization. This indicates a high stability of obtained supramolecular dimers due to the formation of three-centered H-bonding involving a carbonyl O20-atom, a hydroxyl O13-atom and a methoxy O-atom disposed at the ortho position, respecting the OH-group in arylidene moiety.

In terms of crystal packing, both compounds **1** and **2** form the parallel 2D molecular sheets due to π-stacking between the dimers, as shown in Figure 3a,b. Conversely, for compound **1**, all molecules located in the sheets display parallel orientation, the molecules of compound **2** belonging to the neighboring sheets form the angle equal to 81°, as presented in Figure 3b.

The supramolecular motif is completely different when the pyrimidyl N8 atom is involved in H-bonding instead of the carbonyl O8 atom. For compound **3**, the formation of the 1D H-bonded heterochiral chain is observed instead of the supramolecular dimers (see Figure 4) (d_O11…N8_ = 2.763(1) Å). Presumably, it can be explained by the relatively increased basicity of N8 atom caused by the *n*–π interaction of the electron pair belonging to the Ome group disposed in the *orto*-position of the phenyl substituent at the C5-atom with the π-conjugated pyrimidyl moiety displaying the O19…C3N2 centroid distance equal to 2.674 Å (see Appendix A).

In crystal, 1D-heterochiral chains of compound **3** are arranged in a parallel fashion, forming 2D layers along the crystallographic plane (002) due to the π–π interaction between the pyrimidine and 2-arylmethylidene groups (dC9–C13 = 3.341 Å) (Appendix A). It is interesting to note that switching of the H-acceptor from the carbonyl O-atom to the pyrimidyl N-atom earlier led to conglomerate crystallization for similar compounds [15].

Surprisingly, it was revealed that the crystal of the phenyl derivative **4** is composed of homochiral 1D chains displaying a similar connectivity pattern as for compound **3**. It was found that the crystal of **4** consists of only one enantiomer emerging the crystal homochirality (chiral space group P2_1_). The distance between the N8 H-acceptor and the OH H-donor atoms is found to be equal to 3.002 Å in the crystal of **4** (Figure 5). Upon the crystal packing, the homochiral chains are π-stacked in a parallel fashion, forming 2D sheets (see Appendix A) along the (001) crystallographic plane (centroid-centroid distances 3.507 Å).

Halogen bonds (XB) are one of the types of non-covalent interactions that have been actively studied in recent years, which, along with hydrogen bonds, metallophilic interactions and π-stacking, are successfully used in crystal chemical design (or so-called crystal engineering), the construction of supramolecular systems and the creation of materials with controlled properties [17,18,19,20,21,22]. They play an important role in medical chemistry and biochemistry, since the formation of XB is one of the key processes in the metabolism of a number of iodine-containing human hormones and artificial halogen-containing pharmacological preparations [23,24,25]. XB also play a significant role in a number of catalytic reactions, and their formation affects a number of synthetic processes [26]. Recent practical applications based on the phenomenon of XB formation include the stabilization of explosives [27] and the molecular design of compounds with variable photophysical properties [28,29].

In order to study the cooperative influence of halogen bonding and hydrogen bonding on the supramolecular architectures formation in the crystalline phase, thiazolo[3,2-*a*]pyrimidine derivatives–compounds **5** and **6**, containing a Br-atom in a meta- or para-position of phenyl substituent at the C5-atom were synthesized. The crystallization of **5** and **6** upon the slow evaporation conditions at room temperature (see experimental part) led to the formation of the monocrystals suitable for the X-ray diffraction analysis, revealing the H-bonded racemic supramolecular dimers formation similar to those observed for **1** and **2** (see Figure 2) (d_O4…O11_ = 2.795 Å and 2.700 Å for **5** and **6**, respectively) (Figure 6). 

However, the appearance of the Br-atom at the C5 phenyl substituent led to the generation of 1D halogen-bonded chains, where primary formed hydrogen bonded dimers of **5** and **6**, acting as secondary molecular building blocks, are linked by halogen bonding between the pymidyl N-atom (or ester O-atom) and the phenyl Br-atom (d_N8…Br24_ = 3.036 Å and d_Br25…O24_ = 3.120 Å for **5** and **6**, respectively) (see Figure 7, Appendix A). Thus, the X-ray diffraction study of compounds **5** and **6** showed that the Br-atom located on the phenyl substituent at the C5 atom of the thiazalopyrimidine central ring can act as relatively strong XB donor.

In the crystal of compound **5**, all the molecules of the heterocycles provide a Br-atom to form halogen bonds with the pyrimidyl N-atom (see Figure 8a), whereas for compound **6**, only of 50% of the molecules are found to be involved in halogen bonding, which is caused by the convergent disposition of the bonding sites (the Br atom and the O atom from the ester group). In other words, the 1D halogen bonded chain of **6** is composed of two types of racemic dimers: the first one acts as a *bis*-XB-donor due to two Br-atoms belonging to H-bonded molecules in the dimer structure and the other one behaving as a *bis*-XB-acceptor offering two carbonyl ester O-atoms for interaction with the Br-atoms (see Figure 8b and Appendix A). Moreover, it is worth noting that in the crystals of **5** and **6**, the halogen bonding is observed between only the R- or S -isomers, whereas both of the enantiomers participate in the formation of H-bonded supramolecular dimers. 

It was discovered that the changing of the Br-atom position in the phenyl substituent at the C5 atom also influences the crystalline assembly of the obtained halogen and hydrogen bonded supramolecular chains. In the case of compound **5**, containing a *meta*-bromophenyl fragment, the chains are arranged into parallel π-stacked 2D layers (Figure 7a) along the (100) plane (see Appendix A) (centroid-centroid distances 3.523 Å). In contrast to **5**, the π-stacking between the chains in the crystal of **6** leads to 3D molecular network formation (see Appendix A) (centroid-centroid distances 3.641 Å).

For compounds **1**–**6**, the Hirshfield surface analysis was also performed to evaluate the various intermolecular interactions presenting in their crystals (see Appendix A). The generated two-dimensional fingerprint plots illustrated a major contribution of H…H, H…O/O…H and H…C/C…H interactions in the formation of the crystal packings for all of the obtained compounds (see Appendix A, Appendix A).

Since the thiazaolopyrimidine derivatives are good candidates for use as antitumor agents, the cytotoxicity of the obtained compounds **1**–**6** against a series of tumor cell lines with normal Chang liver cells were studied in comparison with the widely used reference drug *Sorafenib* (Table 1).

The cytotoxicity study revealed that compounds **1** and **3** containing an acetyl group and ethoxycarbonyl at the C_6_ atom and *para*-tolyl moiety and *ortho*-methoxyphenyl at the C_5_ atom, respectively, exhibit moderate activity against PC3 (prostate adenocarcinoma cell line) as well as normal cell lines. Moderate cytotoxicity of compound **5** towards both human duodenal adenocarcinoma (Hutu 80) and human liver cells (Chang liver) was also observed. Thiazolopyrimidine derivative **2** containing a *meta*-nitrophenyl fragment at the C_5_ atom was established to be the leading compound in the studied series. This compound showed high cytotoxicity against M-HeLa (epithelioid carcinoma of the cervix) and demonstrated low biological activity on Chang liver (human liver cells). Moreover, it is worth noting that compound **2** displayed efficiency of cytotoxicity towards the M-HeLa cancer cell line two times higher when compared to the reference drug *Sorafenib*, which can be related to the bonding with the specific molecular target in this type of tumor cell. Since the selectivity of compounds acting against tumor cells is an important criterion for assessing cytotoxic action, the selectivity index (SI) as the ratio between the IC_50_ value for normal cells and the IC_50_ value for tumor cells for compound **2** and *Sorafenib*, with respect to the M-HeLa cell line, were calculated. The selectivity index of compound **2** was found to be higher than **6**, whereas the SI for *Sorafenib* in this line was lower than **1**. Therefore, one may conclude the higher SI obtained for compound **2** makes it a consideration as a prospective anti-tumor agent against M-Hella tumor cell lines.

## 3. Materials and Methods

### 3.1. Synthesis and Characterisation

All reagents (Acros Organics (Geel, Belgium), Alfa Aesar (Haverhill, MA, USA)) were used without further purification. The 1,2,3,4-tetrahydropyrimidine−2-thions [11,30,31,32,33], thiazolo[3,2-*a*]pyrimidines [16,34] and ethyl (*Z*)−2-(2-hydroxy−3-methoxybenzylidene)−7-methyl−3-oxo−5-phenyl−2,3-dihydro−5*H*-thiazolo[3,2-*a*]pyrimidine−6-carboxylate **4** [35] were synthesized according the reported methods.

The NMR experiments were performed on a Bruker Avance instrument with an operating frequency of 500 MHz for the recorded ^1^H and ^13^C NMR spectra. The chemical shifts were determined relative to the signals of the residual protons of the CDCl_3_ or DMSO-d_6_ solvents.

The IR spectra in the KBr tablets were recorded on a Bruker Vector−22.

The electrospray ionization (ESI) mass spectra were obtained using a Bruker AmaZon X ion trap mass spectrometer. The melting points were determined on a BOETIUS melting table with an RNMK 05 imaging device.

#### 3.1.1. General Method for Compounds **1**–**6** Preparation

The hydrochloride of the appropriate thiazolo[3,2-*a*]pyrimidine (1 mol) was mixed with a CHCl_3_ (50 mL) and water (50 mL) solution containing 1 mol of NaOH and stirred for 30 min at room temperature. Then, 2-hydroxy−3-methoxybenzaldehyde (1 mol) and a catalytic amount (several drops) of pyrrolidine were added. The resulting mixture was stirred for 3 h under refluxing conditions. After cooling, the formed precipitate was filtered off, washed with ethanol and purified by recrystallization from methanol, followed by drying in vacuum for 1 h at 100 °C temperature, affording a pure product.

6-Acetyl-(Z)−2-(2-hydroxy−3-methoxybenzylidene)−7-methyl−5-(*p*-tolyl)−2*H*-thiazolo[3,2-*a*]pyrimidine−3(5*H*)-one **1**. Yield 97%, orange powder, mp 209–211 °C. IR (KBr, cm^−1^): 3416 (OH); 1706 (C = O); 1547; 1543; 1164; 723. ^1^H NMR (500 MHz, DMSO-d_6_, 25 °C) δ_H_ ppm: 2.23 (s, 3H, CH_3_), 2.35 (s, 3H, CH_3_), 3.82 (s, 3H, OCH_3_), 6.13 (s, 1H, CH-Ar), 6.91–6.97 (m, 2H, CH (Ar)), 7.07–7.09 (m, 1H, CH (Ar)), 7.13–7.15 (m, 2H, CH (Ar)), 7.18–7.20 (m, 2H, CH (Ar)), 7.98 (s, 1H, C = CH), 9.86 (br.s, 1H, CH (Ar)). ^13^C NMR (100 MHz, DMSO-d_6_, 25 °C) δ_C_ ppm: 21.65, 24.20, 31.62, 55.40, 57.01, 80.11, 115.29, 118.58, 119.64, 120.73, 120.79, 127.33, 128.55, 129.40, 130.35, 137.98, 139.14, 147.64, 149.05, 150.24, 153.11, 156.22, 165.55, 197.92. MS (ESI), *m*/*z*, [M + H]^+^: calcd. for C_24_H_23_N_2_O_4_S ^+^: 435,14; found: 435,17 (see Appendix A).

Ethyl (*Z*)−2-(2-hydroxy−3-methoxybenzylidene)−7-methyl−5-(3-nitrophenyl)−3-oxo−2,3-dihydro−5*H*-thiazolo[3,2-*a*]pyrimidine−6-carboxylate **2**. Yield 95%, orange powder, mp 206–208 °C. IR (KBr, cm^−1^): 3465 (OH); 1707 (C = O); 1590; 1543; 1158; 711. ^1^H NMR (500 MHz, DMSO-d_6_, 25 °C) δ_H_ ppm: 1.11 (t, *J* = 7.1 Hz, 3H, OCH_2_CH_3_), 2.42 (s, 3H, CH_3_), 3.83 (s, 3H, OCH_3_), 4.00–4.06 (m, 2H, OCH_2_CH_3_), 6.17 (s, 1H, CH-Ar), 6.91–6.99 (m, 2H, CH (Ar)), 7.09–7.11 (m, 1H, CH (Ar)), 7.66–7.70 (m, 1H, CH (Ar)), 7.77–7.79 (m, 1H, CH (Ar)), 7.99 (s, 1H, C = CH), 8.12 (s, 1H, CH (Ar)), 8.17–8.19 (m, 1H, CH (Ar)), 9.82 (br.s, 1H, OH). ^13^C NMR (100 MHz, DMSO-d_6_, 25 °C) δ_C_ ppm: 14.78, 23.63, 55.55, 57.03, 61.28, 108.46, 115.42, 119.41, 120.73, 120.90, 120.97, 123.47, 124.48, 129.80, 131.49, 135.19, 143.28, 147.72, 148.69, 149.08, 151.44, 165.57. MS (ESI), *m*/*z*, [M + H]^+^: calcd. for C_24_H_22_N_3_O_7_S^+^: 496,12; found: 496,16 (see Appendix A).

Ethyl (*Z*)-2-(2-hydroxy-3-methoxybenzylidene)-7-methyl-5-(2-methoxyphenyl)-3-oxo-2,3-dihydro-5*H*-thiazolo[3,2-*a*]pyrimidine-6-carboxylate **3**. Yield 96%, orange powder, mp 193–195 °C. IR (KBr, cm^−1^): 3400 (OH); 1704 (C = O); 1600; 1545; 1163; 752. ^1^H NMR (500 MHz, DMSO-d_6_, 25 °C) δ_H_ ppm: 1.14 (t, *J* = 7.1 Hz, 3H, OCH_2_CH_3_), 2.29 (s, 3H, CH_3_), 3.72 (s, 3H, OCH_3_), 3.83 (s, 3H, OCH_3_), 3.99–4.05 (m, 2H, OCH_2_CH_3_), 6.16 (s, 1H, CH-Ar), 6.91–6.95 (m, 2H, CH (Ar)), 6.97–7.01 (m, 2H, CH (Ar)), 7.07–7.09 (m, 1H, CH (Ar)), 7.25–7.28 (m, 2H, CH (Ar)), 7.91 (s, 1H, C = CH), 9.70 (br.s, 1H, OH). ^13^C NMR (100 MHz, DMSO-d_6_, 25 °C) δ_C_ ppm: 14.88, 23.37, 54.11, 56.59, 57.02, 60.89, 107.98, 113.05, 115.12, 119.76, 120.64, 120.96, 121.21, 128.33, 130.79, 131.86, 147.57, 149.57, 151.65, 156.63, 158.63, 165.38, 166.21. MS (ESI), *m*/*z*, [M-H]^-^: calcd. for C_25_H_23_N_2_O_6_S^-^: 479,13; found: 479,18 (see Appendix A).

Ethyl (*Z*)−2-(2-hydroxy−3-methoxybenzylidene)−7-methyl−5-phenyl−3-oxo−2,3-dihydro−5*H*-thiazolo[3,2-*a*]pyrimidine−6-carboxylate **4**. Yield 96%, orange powder, mp 231–233 °C. IR (KBr, cm^−1^): 3416 (OH); 1706 (C = O); 1547; 1543; 1164; 723. ^1^H NMR (500 MHz, DMSO-d_6_, 25 °C) δ_H_ ppm: 1.12 (t, *J* = 7.1 Hz, 3H, OCH_2_CH_3_), 2.38 (s, 3H, CH_3_), 3.83 (s, 3H, OCH_3_), 4.02–4.08 (m, 2H, OCH_2_CH_3_), 6.04 (s, 1H, CH-Ar), 6.90–6.99 (m, 2H, CH (Ar)), 7.08–7.11 (m, 1H, CH (Ar)), 7.29–7.37 (m, 5H, CH (Ar)), 7.99 (s, 1H, C = CH), 9.79 (br.s, 1H, OH). ^13^C NMR (100 MHz, DMSO-d_6_, 25 °C) δ_C_ ppm: 14.87, 23.42, 55.86, 57.02, 61.13, 109.55, 115.32, 119.60, 120.69, 120.82, 121.04, 128.38, 129.39, 129.51, 129.69, 141.45, 147.69, 149.07, 152.31, 156.99, 165.54, 165.89. MS (ESI), *m*/*z*, [M-H]^-^: calcd. for C_24_H_21_N_2_O_5_S^-^: 449,12; found: 449,17 (see Appendix A).

Ethyl 5-(3-bromophenyl)-(*Z*)−2-(2-hydroxy−3-methoxybenzylidene)−7-methyl−3-oxo−2,3-dihydro−5*H*-thiazolo[3,2-*a*]pyrimidine−6-carboxylate **5**. Yield 97%, orange powder, mp 190–192 °C. IR (KBr, cm^−1^): 3385 (OH); 1699 (C = O); 1555; 1480; 1161; 715. ^1^H NMR (500 MHz, DMSO-d_6_, 25 °C) δ_H_ ppm: 1.14 (t, *J* = 7.2 Hz, 3H, OCH_2_CH_3_), 2.40 (s, 3H, CH_3_), 3.84 (s, 3H, OCH_3_), 4.01–4.12 (m, 2H, OCH_2_CH_3_), 6.02 (s, 1H, CH-Ar), 6.92–6.96 (m, 1H, CH (Ar)), 6.98–7.00 (m, 1H, CH (Ar)), 7.09–7.13 (m, 1H, CH (Ar)), 7.29–7.37 (m, 2H, CH (Ar)), 7.47–7.49 (m, 1H, CH (Ar)), 7.51–7.54 (m, 1H, CH (Ar)), 8.00 (s, 1H, C = CH), 9.83 (br.s, 1H, OH). ^13^C NMR (100 MHz, DMSO-d_6_, 25 °C) δ_C_ ppm: 14.83, 23.52, 55.51, 57.03, 61.22, 108.86, 115.40, 119.43, 120.71, 120.89, 121.00, 122.52, 127.42, 129.68, 131.52, 132.12, 132.42, 143.87, 147.73, 149.08, 152.88, 157.12, 165.53, 165.68. MS (ESI), *m*/*z*, [M + H]^+^: calcd. for C_24_H_22_BrN_2_O_5_S^+^: 530,41; found: 531,08 (see Appendix A).

Ethyl 5-(4-bromophenyl)-(*Z*)−2-(2-hydroxy−3-methoxybenzylidene)−7-methyl−3-oxo−2,3-dihydro−5*H*-thiazolo[3,2-*a*]pyrimidine−6-carboxylate **6**. Yield 98%, orange powder, mp 234–236 °C. IR (KBr, cm^−1^): 3386 (OH); 1710 (C = O); 1546; 1543; 1156; 718. ^1^H NMR (500 MHz, DMSO-d_6_, 25 °C) δ_H_ ppm: 1.13 (t, *J* = 7.1 Hz, 3H, OCH_2_CH_3_), 2.39 (s, 3H, CH_3_), 3.84 (s, 3H, OCH_3_), 4.02–4.08 (m, 2H, OCH_2_CH_3_), 6.02 (s, 1H, CH-Ar), 6.92–6.96 (m, 2H, CH (Ar)), 6.92–6.99 (m, 2H, CH (Ar)), 7.09–7.10 (m, 1H, CH (Ar)), 7.27 (d, *J* = 8.5 Hz, 2H, CH (Ar)), 7.56 (d, *J* = 8.5 Hz, 2H, CH (Ar)), 8.00 (s, 1H, C = CH), 9.81 (br.s, 1H, OH). ^13^C NMR (100 MHz, DMSO-d_6_, 25 °C) δ_C_ ppm: 14.35, 22.97, 54.88, 56.50, 60.68, 108.47, 114.84, 118.94, 120.18, 120.31, 120.48, 122.22, 129.03, 130.23, 132.10, 140.20, 147.19, 148.55, 152.16, 156.49, 165.00, 165.22. MS (ESI), *m*/*z*, [M + H]^+^: calcd. for C_24_H_22_BrN_2_O_5_S^+^: 530,41; found: 531,07 (see Appendix A).

#### 3.1.2. Crystallization Conditions

The crystals of **1**, **3**, **4** and **6** suitable for an X-ray diffraction study were obtained by slow evaporation of an ethanol solution (30 mL) containing 0.02 mol of the dissolved compound after 7 days.

For compounds **2** and **5**, the crystals were grown by slow evaporation of DMSO or ethanol solutions (15 mL) containing 0.02 mol of the dissolved compounds after 10 days. It was shown that the changing of the crystallization conditions in these cases does not affect the crystal structures of compounds **2** and **5**.

The X-ray diffraction study of **3** and **4** was carried out at the “Belok/XSA” beamline of the Kurchatov Synchrotron Radiation Source [36,37]. The diffraction patterns were collected using a Mardtb goniometer (marXperts GmbH, Werkstraße 3, 22844 Norderstedt, Germany) equipped with a Rayonix SX165 CCD (Rayonix LLC., 1880 Oak Ave UNIT 120, Evanston, IL 60201, USA) 2D positional sensitive CCD detector (λ = 0.7450 Å, φ-scanning in 1.0° steps). All data were collected at 100(2) K.

X-ray diffraction analysis of **1**, **2**, **5** and **6** was performed on a Bruker D8 QUEST automatic three-circle diffractometer with a PHOTON III two-dimensional detector and an IμS DIAMOND microfocus X-ray tube (λ [Mo Kα] = 0.71073 Å) at cooling conditions. Data collection and processing of diffraction data were performed using an APEX3 software package.

All of the structures were solved by the direct method using the SHELXT program [38] and refined by the full-matrix least squares method over F^2^ using the SHELXL program [39]. All of the calculations were performed in the WinGX software package [40], the calculation of the geometry of the molecules and the intermolecular interactions in the crystals was carried out using the PLATON program [41] and the drawings of the molecules were performed using the ORTEP−3 [40] and MERCURY [42] programs.

The non-hydrogen atoms were refined in the anisotropic approximation. The positions of the hydrogen atoms H(O) were determined using difference Fourier maps, and these atoms were refined isotropically. The remaining hydrogen atoms were placed in geometrically calculated positions and included in the refinement in the “riding” model. The crystallographic data of structures **1–6** were deposited at the Cambridge Crystallographic Data Center and the registration numbers and the most important characteristics are given in Table 2.

### 3.2. Hirshfeld Surface Analysis

The 3D Hirshfeld surfaces and 2D fingerprint plots were calculated using Crystal Explorer 17 [43]. The Hirshfeld surfaces were plotted over d_norm_ in the range from −0.5327 (red) to 1.5082 (blue) a.u. (for **1**); from −0.6077 (red) to 1.2545 (blue) a.u. (for **2**); from −0.6304 (red) to 1.2963 (blue) a.u. (for **3**); from −0.3044 (red) to 1.1994 (blue) a.u. (for **4**); from −0.5416 (red) to 1.5037 (blue) a.u. (for **5**); and from −0.5289 (red) to 1.4484 (blue) a.u. (for 6).

### 3.3. Biological Study

#### 3.3.1. Cells and Materials

For the experiments, the tumor cell cultures of M-HeLa clone 11 (epithelioid carcinoma of the cervix, subline HeLa., clone M-HeLa), HuTu 80, human duodenal adenocarcinoma, MCF7—human breast adenocarcinoma (pleural fluid) collected from the Institute of Cytology, Russian Academy of Sciences (St. Petersburg, Russia); PC3—prostate adenocarcinoma cell line collected from ATCC (American Type Cell Collection, Manassas, VA, USA; CRL 1435; human liver cells (Chang liver) and the Research Institute of Virology of the Russian Academy of Medical Sciences (Moscow, Russia) were used for cytotoxicity analysis.

#### 3.3.2. MTT Assay

The cytotoxic effect on cells was determined by the colorimetric method of cell proliferation—MTT test [44]. The cells were seeded on a 96-well Nunc plate at a concentration of 5 × 103 cells per well in a volume of 100 µL of medium and cultured in a CO_2_ incubator at 37 °C until a monolayer was formed. Then, the nutrient medium was removed and 100 μL of the test drug solution in these dilutions was added to the wells prepared directly in the nutrient medium with the addition of 5% DMSO to improve solubility. After 48 h of cell incubation with test compounds, the nutrient medium was removed from the plates and 100 μL of serum-free nutrient medium with MTT at a concentration of 0.5 mg/mL was added and incubated for 4 h at 37 °C. C. Formazan crystals were added to 100 µL of DMSO. The optical density was recorded at 540 nm on an Invitrologic microplate reader (Novosibirsk, Russia). The experiments for all compounds were repeated three times.

## 4. Conclusions

Synthesis of new 2-(2-hydroxy−3-methoxybenzylidene)thiazolo[3,2-*a*]pyrimidine derivatives **1–6** containing phenyl, *m*-nitrophenyl, *p*- and *m*-bromophenyl, *o*-anisyl and *p*- tolyl substituents at the C5 atom was successfully performed in quantitative yields. The single crystal X-ray diffraction study revealed that the supramolecular motif can be controlled by adjusting the interplay between various types of non-covalent interactions, such as hydrogen- or halogen bondings and π–stacking through the rational choice of the substituents at the C2, C5 and C6 atoms in the thiazolopyrimidine scaffold (see Figure 9). It was demonstrated that hydrogen bonding is a particularly efficient tool to switch supramolecular self-assemblies of thiazolopyrimidine derivatives in the crystalline phase from 0D dimers or 1D molecular chains. Thus, the H-bonding between the OH-group and the carbonyl O atom leads to racemic dimers, whereas the formation of polymeric 1D chains is observed when the pyrimidine N-atom is involved in the intermolecular H-bonding between the thiazolopyrimidine species bearing the H-donor group at the substituent at the C2 arylidene fragment. Moreover, it should be mentioned that, depending on the aromatic substituent at the C5 atom, formation of 1D supramolecular heterochiral chains (compounds **3**) or conglomerate crystallization can be realized (compound **4**). One may conclude that H-bonding plays a key role in the chiral discrimination of 2-hydroxy−3-methoxybenzylidene derivatives of thiazolo[3,2-*a*]pyrimidine.

The halogen bonding was revealed to be another important supramolecular synthon, which determines the crystalline organization of 2-hydroxy−3-methoxybenzylidene derivatives of thiazolo[3,2-*a*]pyrimidine. When Br-substituted compounds **5** and **6** were used, it afforded the 1D supramolecular homochiral chains resulting from the halogen-bonding between the H-bonded racemic dimers. It was found that change in the position of the Br atom (meta or para position) in the phenyl substituent at the C5 atom (compounds **5** and **6**) also influences the crystalline assembly of the obtained heterocycles: 2D or 3D π-stacked supramolecular architecture can be generated, respectively.

The cytotoxicity of all of the obtained thiazolopyrimidine derivatives was studied against different tumor cells lines. Thiazolopyrimidine **2** containing a *meta*-nitrophenyl fragment at the C5 atom was established to be the leading compound in the studied series. This compound showed low biological activity towards Chang liver (human liver cells) and high cytotoxicity against M-HeLa (epithelioid carcinoma of the cervix), exceeding twice the efficiency compared to the reference drug *Sorafenib*, which requires further investigation for use as an anti-tumor agent. The synthesis of new thiazolopyrimidines bearing other halogen and hydrogen bonding donor and acceptor functional groups and possessing high cytotoxicity towards tumor cells is in progress.

## Data Availability

The data presented in this study are contained within the article or in the Appendix A, or are available on request from the corresponding author, Svetlana Solovieva.

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
