# Peer review of "(2-Hydroxy-3-Methoxybenzylidene)thiazolo[3,2-a]pyrimidines: Synthesis, Self-Assembly in the Crystalline Phase and Cytotoxic Activity"

_ijms, 2023, doi:10.3390/ijms24032084_

Round 1

Reviewer 1 Report

In the submitted manuscript the synthesis, physicochemical and biological analysis of six (2-Hydroxy-3-Methoxybenzylidene)thiazolo[3,2-a]pyrimidines is being reported. The manuscript requires revision, my questions and comments are presented below.

In the introduction a figure should be introduce to present the compounds described in this part. It would be much easier for the reader to simultaneously look at the chemical structures and read about them.

Line 83, I would continue to use the bold for 1-6, similarly as in line 77

Line 92-it should be “bromo”

Lines 95 and 99, it should be  oC”

Lines 83-101, usually those information are part of the “materials and method” section

Line 104, what do you mean by “the structures in solid state are in agreement with those in solution”?

Line 138, it should be “in the Figure”

Lines 221-222, it is questionable whether this is really the “most important” stage. What about stability, solubility etc.?

Lines 224-225, IC50 is a well-known index, it doesn’t need to be explained here

Line 225, not “shooting” but “recording”

Table 1, when compared to Sorafenib actually none of the studied compounds seems to be superior. Could you please comment on this?

In the 6 two conformations in solid state has been found. Could you please comment on the differences among them?

Usually, when comparing such crystal structures, Hirshfeld surface analysis is also performed. In my opinion it is crucial for this work.

Author Response

We are grateful to the reviewers for their constructive remarks, comments and suggestions. We've revised our manuscript according to their recommendations. The changes are highlighted in yellow in the revised version of the manuscript. Below is also our detailed point-by-point response.

Comments and Suggestions from Reviewer 1.

In the submitted manuscript the synthesis, physicochemical and biological analysis of six (2-Hydroxy-3-Methoxybenzylidene)thiazolo[3,2-a]pyrimidines is being reported. The manuscript requires revision, my questions and comments are presented below.

  • In the introduction a figure should be introduce to present the compounds described in this part. It would be much easier for the reader to simultaneously look at the chemical structures and read about them.

Figure 1 has been added to the Introduction section of the revised version of the manuscript.

  • Line 83, I would continue to use the bold for 1-6, similarly as in line 77
  • Line 92-it should be “bromo”
  • Lines 95 and 99, it should be  “oC”
  • Line 138, it should be “in the Figure”
  • Lines 224-225, IC50 is a well-known index, it doesn’t need to be explained here
  • Line 225, not “shooting” but “recording”

The manuscript was carefully revised in accordance with referee remarks.

  • Lines 83-101, usually those information are part of the “materials and method” section

We consider this part concerning discussion about particularities of synthetic reactivity of presented compounds important for this work. In this respect, it was placed into the Results and Discussion section.

  • Line 104, what do you mean by “the structures in solid state are in agreement with those in solution”?

A phrase “The structures in solid state are in agreement with those in solution” means that no differences between the structures of dissolved compounds which were studied in solution by NMR-spectroscopies, MALDI-TOF mass-spectrometry and the solid-state structures established by single crystal X-ray diffraction are observed and no structural changing such as, for example, cis-trans isomerism of the double C=C bond connected to arylidene moiety, occurs when compounds are dissolved.

  • Lines 221-222, it is questionable whether this is really the “most important” stage. What about stability, solubility etc.?

We would like to say that the study of cytotoxicity of obtained thiazolopyrimidines in this work play the crucial role in the investigation of their biological activity and allows to evaluate the perspectives of their further application as anti-tumor drugs. The synthesized derivatives are stable under normal conditions in solution and in solid-state after exposing to the air for a long time which was attested by NMR, IR- spectroscopies, mass-spectrometry data. These derivatives demonstrate a good solubility in alcohols (MeOH, EtOH) and DMSO.

  • Table 1, when compared to Sorafenib actually none of the studied compounds seems to be superior. Could you please comment on this?

As it was attested within the series of studied compounds, compound 2 display a particular cytotoxic activity with respect the M-HeLa cell line. Its efficiency is found to be practically in two times higher than this one for widely used reference drug Sorafenib (IC50 = 16.2±1.3 µM and 25.0 ± 1.8 µM for 2 and Sorafenib, respectively). In addition, another very important point is that the selectivity index against cancer cells M-HeLa tumor cells calculated for compound 2 is higher than 6 whereas for Sorafenib it is lower than 1 that both with a low toxicity effect with respect of normal cell lines (Chang liver cells HeLa) offers the additional advantage to compound 2 to be potentially used as selective anti M-Hela tumor cells agent.

  • In the 6 two conformations in solid state has been found. Could you please comment on the differences among them?

In the solid state, there is not two conformations were found, but two independent thiazolopyrimidine molecules within the unit cell of 6 which are different by the involvement of the Br- or O-ester- atoms into the 1D halogen-hydrogen bonded chain formation. Whereas the first one acts as XB-donor due to two presence of Br-atoms, the another one behaves as XB-acceptor offering carbonyl ester O-atom for interaction with Br-atoms of the fist one. In order to clarify the this difference the appropriate figures S28 were added to Supporting information file.

  • Usually, when comparing such crystal structures, Hirshfeld surface analysis is also performed. In my opinion it is crucial for this work.

The calculated 3D Hirshfeld surfaces as well as 2D fingerprint plots have been added to the revised version of the Supplementary material file (see Figures S31, S32 and Table S5). Some appropriate comments on the contribution of the various kind of interactions in the crystal packings are also added in the main text of the manuscript.

Reviewer 2 Report

In this manuscript, Solovieva and co-authors synthesized a series of new compounds based on 2-hydroxy-3-methoxybenzylidenethiazolo[3,2-a]pyrimidine moieties. They characterized the compounds with various techniques, including H/C NMRs, IR, MS, and XRD. The self-assembly properties in their crystalline phase were interpreted in detail. Finally, the cytotoxicity of these novel compounds was evaluated with an MTT assay toward different types of cell lines and demonstrated that compound 2 exhibits high efficiency against cervical adenocarcinoma cells and minimal cytotoxicity toward normal cells. The experiments are well-designed, and the results are clearly presented. The reviewer finds it suitable to be published in IJMS after minor revision: 

1 The reaction conditions listed for the first step in Scheme 1 are confusing. It is hard to understand without carefully reading the main text's description. Thus, the reviewer recommends that the authors change the conditions presented to make it easier for the readers to understand. Also, the authors need to show the yields for each intermediate on the scheme. 

2 The authors need to go through the manuscript carefully and fix multiple compound # that are not bold. (i.e., Line 83, Figure 4 heading, Table 1, etc.)

3 It is interesting that the compounds reported herein show significantly different IC50 values towards other cell lines. It is recommended to provide some reasons behind this. 

4 For NMRs in SI, it is desired to show integrations of the peaks in the zoomed-in spectra. 

5 For MS reports, please provide the calculated values as well. Also, since compounds 3 and 4 are tested under the negative mode, they should be reported as [M-H]- instead of [M-H]+. 

Author Response

We are grateful to the reviewers for their constructive remarks, comments and suggestions. We've revised our manuscript according to their recommendations. The changes are highlighted in yellow in the revised version of the manuscript. Below is also our detailed point-by-point response.

Comments and Suggestions from Reviewer 2

In this manuscript, Solovieva and co-authors synthesized a series of new compounds based on 2-hydroxy-3-methoxybenzylidenethiazolo[3,2-a]pyrimidine moieties. They characterized the compounds with various techniques, including H/C NMRs, IR, MS, and XRD. The self-assembly properties in their crystalline phase were interpreted in detail. Finally, the cytotoxicity of these novel compounds was evaluated with an MTT assay toward different types of cell lines and demonstrated that compound 2 exhibits high efficiency against cervical adenocarcinoma cells and minimal cytotoxicity toward normal cells. The experiments are well-designed, and the results are clearly presented. The reviewer finds it suitable to be published in IJMS after minor revision:

  • The reaction conditions listed for the first step in Scheme 1 are confusing. It is hard to understand without carefully reading the main text's description. Thus, the reviewer recommends that the authors change the conditions presented to make it easier for the readers to understand. Also, the authors need to show the yields for each intermediate on the scheme.

  • The authors need to go through the manuscript carefully and fix multiple compound # that are not bold. (i.e., Line 83, Figure 4 heading, Table 1, etc.)

  • For NMRs in SI, it is desired to show integrations of the peaks in the zoomed-in spectra.

  • For MS reports, please provide the calculated values as well. Also, since compounds 3 and 4 are tested under the negative mode, they should be reported as [M-H]- instead of [M-H]+.

We did our best to improve the text and spectra according to the referee remarks. The appropriate corrections were made all along the manuscript and Supporting Information file.

  • It is interesting that the compounds reported herein show significantly different IC50 values towards other cell lines. It is recommended to provide some reasons behind this.

In our previous work, it was shown that similar compounds (2-(2- / 4- hydroxybenzylidene)thiazolo[3,2-a]pyrimidines) had demonstrated cytotoxic effects against the M-HeLa and HuTu80 lines which was associated with the induction of the mitochondrial apoptosis pathway and cell cycle delay in the G1/G0 phase. [Agarkov A.S., Nefedova A.A., Gabitova E.R., Ovsyannikov A.S., Amerhanova S.K., Lyubina A.P., Voloshina A.D., Dorovatovskii P.D., Litvinov I.A., Solovieva S.E., Antipin I.S. Synthesis, self-assembly in crystalline phase and anti-tumor activity of 2-(2-/4-hydroxybenzylidene)thiazolo[3,2-a]pyrimidines // Molecules. – 2022. – Vol. 27, Is. 22. Art. 7747. DOI: 10.3390/molecules27227747].

In the case of compound 2, an even higher selectivity was observed with respect to cells of the M-HeLa line, probably due to the inhibition of a specific molecular target of this line, for example, the enzyme telomerase, inhibiting the tumor cells division (reproduction). It is known that in the case of the cervical cancer, the telomerase activity is 100%. HeLa cells with a decreased in telomerase activity also become more susceptible to the effects of a number of chemotherapeutic agents and radiation [Xi L., Chen G., Zhou J., Xu G., Wang S., Wu P., Zhu T., Zhang A., Yang W., Xu Q., Lu Y., Ma D. (2006) Apoptosis. 2006. 11(5). Р. 789-798]. Telomerase activity does not occurs in human somatic cells, so it can explain the absence of cytotoxic action on normal cells.

Round 2

Reviewer 1 Report

The Authors have made the corrections and this version can be accepted for publication.